# Development of the Conceptualization of Pain Questionnaire: A Measure to Study How Children Conceptualize Pain

**DOI:** 10.3390/ijerph18073821

**Published:** 2021-04-06

**Authors:** Isabel Salvat, Cristina Adillón, Eva Maria Andrés, Sonia Monterde, Jordi Miró

**Affiliations:** 1Department of Medicine and Surgery, Faculty of Medicine and Health Sciences, Institut d’Investigació Sanitària Pere Virgili, Universitat Rovira i Virgili, 43204 Reus, Spain; cristina.adillon@urv.cat (C.A.); sonia.monterde@urv.cat (S.M.); 2Grupo de Investigación “Gestión en el Paciente Sangrante” IdiPaz, Department of Economía de la Empresa, Universidad Rey Juan Carlos Economía Aplicada II y Fundamentos Análisis Económico, 28933 Madrid, Spain; e.andres@live.com; 3Unit for the Study and Treatment of Pain-ALGOS, Chair in Pediatric Pain URV-FG, Research Center in Behavior Assessment and Measurement, Department of Psychology, Universitat Rovira i Virgili, 43007 Tarragona, Spain; jordi.miro@urv.cat

**Keywords:** survey, children, health knowledge, chronic pain, pain education

## Abstract

(1) Background: Research has shown that thoughts about pain are important for the management of chronic pain in children. In order to monitor changes in thoughts about pain over time and evaluate the efficacy of treatments, we need valid and reliable measures. The aims of this study were to develop a questionnaire to assess a child’s concept of pain and to evaluate its psychometric properties; (2) Methods: This is a cross-sectional, two-phase, mixed-method study. A total of 324 individuals aged 8 to 17 years old responded to the newly created questionnaire. The Conceptualization of Pain Questionnaire (COPAQ) was calibrated using the Rasch model. The chi-square test was used for the fit statistics. Underfit and overfit of the model were determined and a descriptive analysis of infit and outfit was conducted to identify who responded erratically. Internal consistency was measured using the Person Separation Index (PSI); (3) Results: Fit to the Rasch model was good. Suitable targeting indicated which items were simple to answer; Person Fit identified 9.56% children who responded erratically; PSI = 0.814; (4) Conclusions: The findings suggest that COPAQ is a measure of a child’s concept of pain that is easy to administer and respond to. It has a good fit and a good internal consistency.

## 1. Introduction

Chronic pain is a common and debilitating health problem in children and adolescents [1]. For example, in a study with a community sample of 8- and 16-year-olds, Huguet and Miró showed that 37% of the participants reported chronic pain problems [2]. However, the prevalence rate, which is increasing [3], varies across epidemiological studies [1,2,3,4]. Moreover, children with chronic pain report significant chronic pain-related interferences in their physical, psychological, and social functioning [1,5,6,7].

To manage their pain-related problems, children with chronic pain meet numerous healthcare professionals, and sometimes they have difficulties in expressing their feelings or their thoughts about pain [8,9]. For example, in a qualitative study with children with chronic pain, Dell’Api and colleagues [10] found that children felt misunderstood, did not believe in their interactions with their healthcare professionals, and developed negative perceptions about their pain problems. Importantly, the data also showed that these experiences influenced children’s approaches to future encounters with healthcare professionals. Research has shown that healthcare professionals need to enable children to communicate their feelings and help them understand their experience of pain so that they can adjust and cope better [10].

In adults, studies have shown that patients with pain who are not informed or who are incorrectly informed often consider their pain to be more threatening, have lower pain tolerance thresholds and more catastrophic pain-related thoughts, and use fewer adaptive coping strategies [11]. Robins and colleagues also suggested an association between the quality of the information received and how children cope with and adjust to pain [12].

There is mounting evidence to show that the reconceptualization of pain through education is central to the treatment of adults with chronic pain [13,14,15,16,17,18,19,20,21,22,23,24,25]. In this respect, education strategies to explain pain to patients have proved to be able to change pain-related attitudes [23,26] and catastrophic thinking about pain [27], which in turn can help improve psychological and physical function [18,27,28,29,30]. Pain education has been used in interventions designed to treat individuals with chronic pain [18,19,20,21,22,23,24,25,26,27,31] and is considered a concomitant measure for reducing post-surgery pain [15,16,17,32,33].

Questionnaires have been developed to assess how individuals conceptualize their pain. For example, the revised Neurophysiology of Pain Questionnaire [34] was devised to be used with adults, and has proved to be a useful tool for studying the conceptualization of the biological mechanisms underlying pain, and the effects of educational interventions on patients’ pain experience [20,35,36]. Louw et al. [37] used this questionnaire as the assessment tool for one programme of education in the neurophysiology of pain in children aged 12 years old.

Pate and colleagues pointed out the need for a questionnaire specifically for children [38], and recently developed the Concept of Pain Inventory (COPI) [39] for this purpose. The COPI has fourteen items, all of which were developed according to contemporary pain science, and review from a panel of clinical and research-focused international pediatric pain experts. Although it has shown an acceptable internal consistency (α = 0.78) and test-retest reliability (r = 0.54), it has several limitations. First, each item is rated on a 5-point Likert scale ranging from 0 = “strongly disagree” to 4 = “strongly agree” where higher scores indicate higher levels of alignment with contemporary pain science. All items are written so that the correct answer is always “strongly agree”, which could induce random responses from respondents. In addition, it seems that the COPI might not be suited for all children. For example, 39% of the respondents in the study required parental assistance to complete the questionnaire.

Given these considerations, the aims of this study were to (1) develop a new questionnaire to assess a child’s concept of pain and (2) conduct a preliminary analysis to evaluate its psychometric properties.

## 2. Materials and Methods

### 2.1. Study Design

This cross-sectional, two-phase, mixed-method study was conducted between February 2016 and December 2018 at the School of Medicine and Health Sciences of the Universitat Rovira i Virgili (Reus, Spain). The study adhered to the tenets of the Declaration of Helsinki and received ethical approval from the local institutional review board (Pere Virgili Institute; Ref. CEICm: 114/2018).

Informed consent for participation in the study was obtained from the children and their parents or guardians.

### 2.2. Participants

The participants in this study were a convenience sample of schoolchildren aged 8–17 years from Vedruna Sagrat Cor School (Tarragona, Spain), a school large enough to have a sufficient sample and near to the School of Medicine and Health Sciences. Potential participants were included if were able to read and write in Catalan and excluded if they suffered from an intellectual disability that, according to the teachers, could interfere with their participation in the study procedures. The teachers agreed with the parents or guardians of the participants that they would sign the consent forms, which they did at regular school meetings. After consent had been obtained, the participants also gave their assent to participate immediately before they completed the questionnaire. Both the informed consent and assent were obtained by the researchers in the presence of the teachers.

### 2.3. Procedure

The development and analysis of the new questionnaire (Conceptualization of Pain Questionnaire; COPAQ) consisted of two stages.

Phase 1: Content Development using Qualitative Methods.

In this stage, we developed the item content. The first two authors generated a pool of 17 (True/False) items in order to cover the seven dimensions that an expert survey on the concept of pain had identified: namely, "External influences on pain"; "Learning about pain is helpful"; "Pain and injury are not closely related"; "Pain is about protection"; "How pain works", "Things are always changing in your brain and body"; and "Pain is a conscious experience" [38]. The items were all in Catalan.

Next, an expert Catalan linguist verified whether the items were written in language that was easy to understand by children aged 8–17 years old. On the basis of his suggestions, we revised and slightly modified the wording of nine items so that they were written in the first person (for example, we changed "You can feel pain even when you do not have an injury ‘for’ I can have pain even when I do not have an injury").

This preliminary list of 17 items was administered and pilot-tested on a sample of 23 children aged 8–17 years old to test comprehensibility and identify misunderstandings. Participants in the pilot test were asked to respond to the questionnaire and identify any item they found difficult to understand. Two items seemed a little confusing and repetitive, so they were removed. No other difficulties were observed during the pilot test. The remaining 15 items were included in the Conceptualization of Pain Questionnaire (COPAQ).

Phase 2: Psychometric Assessment of the Questionnaire using Rasch Analysis.

The Conceptualization of Pain Questionnaire (COPAQ) consist of 15 items and asks respondents to respond if they believe the items/statements to be true or false, although they are allowed to respond undecided. A score of 0 is given to incorrect responses and those marked as undecided; the sum of all the correct answers gives the total score. The higher the score, the better the participant understands the concept of pain.

Table 1 lists the items of the COPAQ translated into English along with their correct answers (see Appendix A for the original Catalan version of the questionnaire).

Sample size range was established on the basis that at least 10 observations per response option are needed for item threshold analysis [40] and at least 50 participants are needed to determine item fit with the Rasch model [41]. Therefore, a sample of at least 150 participants should be available. Allowing for the removal of extreme scores, a sample of 243 persons is required for Rasch analysis to ensure item calibration stability within 0.5 logits with 99% confidence [34]

This 15-item questionnaire was administered to schoolchildren who went to the participating school, whose parents had signed the informed consent, and who assented to participate just before the questionnaire administration started.

In the presence of the teachers, the researchers administered the consent forms and the questionnaire. The participants were asked to indicate whether they believed the statements in the questionnaire were true or false. They were also allowed to respond that they were undecided.

The researchers and the teachers checked that the students did not comment on each other’s responses.

### 2.4. Statistical Analyses

We first computed descriptive statistics (means and standard deviations for continuous variables, and numbers and percentages for dichotomous variables) and then we used Rasch analysis to evaluate the internal consistency of the questionnaire. This evaluates whether high scores consistently include correct answers to easy questions and low scores consistently do not include correct answers for harder questions. By looking at the questionnaire as a whole, this analysis provides visual representations of the way individual questions can be rescaled or removed to better fit the data and target the specific criteria the questionnaire seeks to measure [42]. Rasch analysis makes it possible to study whether the level of difficulty of the items matches the respondents’ level of comprehension [42].

The steps involved in Rasch analysis [42,43] are the following:Targeting is evaluated by descriptive analysis of the distribution of the answers to each item and comparison of the summary statistics obtained. In this way, the items are considered too easy if over 95% of the participants respond correctly and too difficult if less than 5% do. If any participant exceeds these thresholds, their answers are eliminated from the analysis.Unidimensionality describes the questionnaire’s ability to measure a single construct. It is best determined by fit statistics (infit and outfit). Both these statistics define how well each item conforms to the underlying construct. The infit statistics are more revealing since they are less sensitive to the effect of the outliers.Fit adjustment of the answers based on the Rasch model is evaluated using the Chi-Square contrast for each item with respect to the general model.The Person-Fit Identification of people with acceptable infit statistics but excessive outfit statistics is considered to indicate careless mistakes.To measure internal consistency, Rasch analysis estimates the Person Separation Index (PSI), which is equivalent to Cronbach’s alpha in other types of analysis. A PSI of less than 80 suggests that an instrument may not be sensitive enough to distinguish between high and low performers [34].

The association of participants’ responses to sex and age was analysed using the Chi-Square test and the non-parametric Mann–Whitney U test, respectively.

Data analyses were performed using SPSS v.25 for Windows. Rasch analysis was performed using R Statistical Software Package (version 3.5.1).

## 3. Results

### 3.1. Participants

Of the 556 possible participating schoolchildren, 324 agreed to participate. None of those were excluded when applying the eligibility criteria, so the resulting sample was 324 schoolchildren aged 8–17 years old (mean = 12.41; SD = 2.80). Most participants were females (56.33%) (see Table 2).

### 3.2. Rasch Analysis

#### 3.2.1. Targeting

None of the participants answered more than 95% or less than 5% of the items correctly. Therefore, the level of difficulty of the questionnaire could be said to be adequate. This result also enabled us to use the whole sample for the Rasch analysis.

#### 3.2.2. Unidimensionality (Item Difficulty)

Rasch analysis divided the answers to the questions into two categories: true or false. Table 3 shows the adjustment statistics for the 15 items. Rasch theory states that the parameters should be interpreted as the ease with which each item is completed (the beta value of the model) so zero is the average value, positive values indicate a difficult question and negative values indicate a simple question. In general, the cut-off points for determining whether an item is a poor fit are 3 and −3. Therefore, any item that exceeds those levels are removed from the questionnaire.

None of the Beta coefficients exceeded the threshold so no items were removed from the questionnaire (see Table 3). Items 2, 5, 7, 8, and 15 can be considered difficult. The remaining items had a negative Beta coefficient, which indicates that they were easier to answer.

#### 3.2.3. Fit Adjustment

The fit adjustment of the answers based on the Rasch model was evaluated by Chi-Square contrast for each item. The Chi-Square test determines whether the item fits the general model. None of the contrasts was statistically significant, which confirms that all the items fit the questionnaire (see Table 4).

By analyzing the main components and focusing on the residual correlation matrix and factorial load matrix, unexpected patterns in the data can be identified. If the standardized residual correlations are high, redundant items should be removed from the questionnaire. In this study, however, some correlations did not reach 0.4.

#### 3.2.4. Person Fit

The True/False nature of the COPAQ makes it susceptible to guesses. However, the Rasch software provides fit statistics that identify children who respond erratically. Acceptable infit statistics but excessive outfit statistics indicate respondents who committed careless mistakes.

To identify respondents who answered randomly or whose responses did not agree with the pattern of the responses to the questionnaire, Rasch analysis provides an individual analysis for each of the 324 children. Thirty-one (9.56%) respondents had a chi-square test p value below 0.05, which shows that their responses did not fit the model of responses to the questionnaire. Their responses can be considered to be random.

#### 3.2.5. Internal Consistency or Reliability

To measure internal consistency, Rasch analysis estimates a parameter known as the Person Separation Index (PSI), which is equivalent to Cronbach’s alpha. In this study, the PSI was 0.81, which indicates good internal consistency for the questionnaire in this sample.

#### 3.2.6. The Association between Concept of Pain, and Sex and Age

When we analyzed the association of responses with sex and age using the chi-square test and the non-parametric Mann–Whitney U test, respectively, we did not find any statistically significant differences: sex: *p* = 0.78; age: *p* = 0.45.

Table 5 shows the scores for the 15 items of the questionnaire stratified by gender. It shows that the distribution of responses was similar for all participants except for three items: item number 3, for which girls answered “undecided” more frequently and therefore had a lower rate of True and False responses (*p* = 0.030); item number 7, for which girls responded "False" more often than boys (*p* = 0.049); and item number 10, for which boys responded "undecided" more than girls (*p* = 0.011).

## 4. Discussion

The aims of this study were to develop a new instrument to assess a child’s concept of pain, the Conceptualization of Pain Questionnaire (COPAQ), and study some of its psychometric characteristics (i.e., dimensionality, adjustment, and internal consistency) when used with a sample of children and adolescents aged 8–17 years old. The data indicated that the COPAQ is a unidimensional measure, with good fit, and a good internal consistency.

Although other measures can be used to assess an individual’s concept of pain, the COPAQ responds to the need for a pediatric version [37]. Importantly, it is quite different from another similar measure (COPI; [39]) in major ways. First, the COPAQ is easy to administer and comprehend. The questionnaire’s overall level of difficulty was suitable, as was that of each item. In this study, participants did not report any difficulties when responding to the items, and did not request any help. However, in the study by Pate and colleagues (2020) conducted with a similar sample of a school-aged group of students, 39% of the participants in the study used parental assistance to complete the questionnaire. Second, a potential problem when children, particularly young ones, respond to questionnaires is that they may respond randomly [44]. The COPI is written in such a way that the correct answer is always “strongly agree”, which could induce random responses from respondents as all the questions have the same answer. In our questionnaire there are true and false alternatives and only 9.56% of responses could be considered random. The data showed no statistically significant associations between responses to the questionnaire and participants’ age or sex.

The study has several limitations, which should be taken into account when interpreting the results. First, our sample was a convenience one, so it may or may not be representative of the population. Future research should use COPAQ with other samples of schoolchildren or in clinical settings to explain the findings, as COPI did. Second, although the data have shown good fit and internal consistency, some important psychometric properties were not examined; including sensitivity to change over time, construct validity, item discrimination and test-retest reliability. Therefore, future studies are warranted to evaluate other properties of the COPAQ.

In addition, future studies should explore the contribution of combining the COPAQ with qualitative intersubjective analysis for assessing changes in the conceptualization of a child’s concept of pain.

## 5. Conclusions

Despite the study’s limitations, the findings suggest that the COPAQ is a measure of a child’s concept of pain that is easy to administer and respond to. Research studying the effects of education or other cognitive strategies on the management of pain in children and adolescents could take advantage of the unique characteristics of the questionnaire. We encourage future investigations to combine the COPAQ with qualitative analysis for assessing changes in the child’s concept of pain.

## Figures and Tables

**Table 1 ijerph-18-03821-t001:** Conceptualization of Pain Questionnaire translated into English.

Items	Answer
1. I only have pain when I am injured or about to be injured.	F
2. If people have pain for a long time, surely they have a problem that cannot be cured.	F
3. When I have pain, it is because my body sends information to the brain.	F
4. If a medication does not take away my pain, the injury is more serious than I thought.	F
5. My brain decides when I have to feel pain.	T
6. The pain I feel depends on the situation I find myself in.	T
7. It is possible to have pain but not realize it.	F
8. When I am injured I am sure I will have pain.	F
9. If someone can be distracted from their pain, it means that they are not experiencing real pain.	F
10. The same injury can produce the same intensity of pain for different people.	F
11. If a pain varies in intensity according to the state of mind, it is a pain that is not real.	F
12. Having pain for a long time means that you will have pain forever.	F
13. I can have pain even when I do not have an injury.	T
14. Sometimes, pain may come from thinking that you have hurt yourself, even if you are well.	T
15. A more serious injury will cause more pain than a less serious injury.	F

T = true; F = false.

**Table 2 ijerph-18-03821-t002:** Distribution of the 324 participants.

School Year	*n* (%)	Mean Age, Years (SD)	*n*, Girls (%)
Year 1	38 (11.73%)	8.29 (0.46)	17 (44.74%)
Year 2	34 (10.49%)	9.21 (0.41)	17 (50.00%)
Year 3	39 (12.04%)	10.37 (0.60)	22 (56.41%)
Year 4	38 (11.73%)	11.22 (0.42)	17 (44.74%)
Year 5	33 (10.19%)	12.26 (0.44)	24 (72.73%)
Year 6	26 (8.02%)	13.11 (0.31)	19 (73.08%)
Year 7	41 (12.65%)	14.14 (0.50)	22 (53.66%)
Year 8	44 (13.58%)	15.33 (0.52)	26 (59.09%)
Year 9	14 (4.32%)	16.30 (0.48)	11 (78.57%)
Year 10	17 (5.25%)	17.29 (0.47)	14 (82.35%)

**Table 3 ijerph-18-03821-t003:** Adjustment statistics for the 15 items.

Item	β (Easiness Parameters)	SE	Correct Responses	95% Confidence Interval	
				Lower Bound	Upper Bound
1	−0.750	0.185	62.96%	−1114	−0.387
2	0.562	0.156	8.64%	0.257	0.868
3	−0.660	0.139	24.38%	−0.933	−0.386
4	−0.527	0.158	43.83%	−0.837	−0.217
5	0.550	0.158	12.65%	0.240	0.861
6	−0.541	0.199	67.28%	−0.931	−0.151
7	0.145	0.164	34.57%	−0.175	0.466
8	0.876	0.189	28.09%	0.506	1.24
9	−0.534	0.183	60.49%	−0.893	−0.174
10	−1.334	0.204	70.06%	−1.734	−0.935
11	−1.646	0.184	59.26%	−2.006	−1.286
12	−1.802	0.354	89.81%	−2.496	−1.108
13	−1.315	0.182	90.43%	−1.672	−0.958
14	−1.168	0.342	61.42%	−1.839	−0.497
15	0.186	0.154	25.62%	−0.115	0.488

SE = standard error.

**Table 4 ijerph-18-03821-t004:** Fit adjustment of the items.

Item	Chi-Square	df	*p* Value	Outfit MSQ	Infit MSQ	Outfit	Infit
1	321.57	323	0.512	0.993	0.993	−0.08	−0.07
2	346.46	323	0.052	1.162	1.029	1.23	0.40
3	303.01	323	0.781	0.935	0.962	−1.04	−0.73
4	296.67	323	0.851	0.916	0.918	−1.41	−1.44
5	286.32	323	0.930	0.884	0.933	−0.99	−0.87
6	320.39	323	0.531	0.989	0.989	−0.12	−0.13
7	329.32	323	0.392	1.016	1.003	0.25	0.07
8	308.47	323	0.710	0.952	0.989	−0.50	−0.11
9	283.14	323	0.956	0.874	0.876	−1.89	−1.85
10	302.94	323	0.782	0.935	0.936	−0.79	−0.78
11	288.92	323	0.914	0.892	0.891	−1.64	−1.65
12	289.70	323	0.908	0.894	0.921	−0.64	−0.48
13	300.29	323	0.813	0.927	0.928	−1.06	−1.03
14	296.18	323	0.855	0.914	0.919	−0.49	−0.47
15	276.73	323	0.971	0.854	0.931	−1.78	−1.10

df = degree of freedom; MSQ = Mean-Squared.

**Table 5 ijerph-18-03821-t005:** Response to the 15 items of the questionnaire stratified by gender.

Item		Men	Women	*p* Value
1	False	83 (61.94%)	118 (64.13%)	0.528
	U	19 (14.18%)	31 (16.85%)	
	True	32 (23.88%)	35 (19.02%)	
2	False	9 (6.72%)	18 (9.78%)	0.278
	U	18 (13.43%)	0 (0.00%)	
	True	107 (79.85%)	32 (72.83%)	
3	False	41 (30.60%)	38 (20.77%)	0.030
	U	31 (23.13%)	65 (35.52%)	
	True	62 (46.27%)	80 (43.72%)	
4	False	63 (47.01%)	77 (41.85%)	0.546
	U	24 (17.91%)	41 (22.28%)	
	True	47 (35.07%)	66 (35.87%)	
5	False	93 (69.40%)	136 (73.91%)	0.275
	U	19 (14.18%)	29 (15.71%)	
	True	22 (16.42%)	19 (10.33%)	
6	False	25 (18.66%)	41 (22.28%)	0.734
	U	16 (11.94%)	21 (11.41%)	
	True	93 (69.40%)	122 (66.30%)	
7	False	42 (31.34%)	69 (37.70%)	0.049
	U	15 (11.19%)	33 (18.03%)	
	True	77 (57.46%)	81 (44.26%)	
8	False	40 (29.85%)	50 (21.17%)	0.685
	U	11 (8.21%)	20 (10.87%)	
	True	83 (61.94%)	114 (61.96%)	
9	False	77 (57.46%)	116 (63.04%)	0.593
	U	21 (15.67%)	24 (13.04%)	
	True	36 (26.87%)	44 (23.91%)	
10	False	93 (69.92%)	130 (71.04%)	0.011
	U	29 (21.80%)	22 (12.02%)	
	True	11 (8.27%)	31 (16.94%)	
11	False	74 (55.22%)	114 (62.30%)	0.407
	U	36 (26.87%)	44 (24.04%)	
	True	24 (17.91%)	25 (13.66%)	
12	False	119 (90.84%)	167 (90.76%)	0.484
	U	9 (6.87%)	9 (4.89%)	
	True	3 (2.29%)	8 (4.35%)	
13	False	8 (6.02%)	6 (3.26%)	0.408
	U	5 (3.76%)	10 (5.43%)	
	True	120 (90.23%)	168 (91.30%)	
14	False	27 (20.15%)	29 (15.76%)	0.273
	U	23 (17.16%)	44 (23.91%)	
	True	84 (62.69%)	111 (60.33%)	
15	False	28 (21.05%)	44 (24.04%)	0.776
	U	23 (17.29%)	33 (18.03%)	
	True	82 (61.65%)	106 (57.92%)	

## Data Availability

The study did not report any data.

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
