# Peer review of "Development of the Conceptualization of Pain Questionnaire: A Measure to Study How Children Conceptualize Pain"

_ijerph, 2021, doi:10.3390/ijerph18073821_

Round 1

Reviewer 1 Report

The authors make a great effort for sistematizing and to measure an actually difficult concept: pain, more in children. The instrument they have articulated is really interesting. Congratulations.

I recommend the authors to incorporate qualitative instruments and intersubjective analysis because a subject like pain requieres this effort.

The originality of the paper is limited by their methodological orientation (this means that there are some methodological limitations for the research, but in terms of the paper, the core logic is strong). This is why i recommend to introduce another qualitative instruments like focus groups or discussion groups, and over all, intersubjective analysis, but not for improving the paper but research.

Author Response

The authors make a great effort for sistematizing and to measure an actually difficult concept: pain, more in children. The instrument they have articulated is really interesting. Congratulations.

Authors’ response: Thank you for the kind comments.

I recommend the authors to incorporate qualitative instruments and intersubjective analysis because a subject like pain requires this effort.

Authors’ response: We agree that qualitative instruments and intersubjective analysis can contribute to this type of study -more so, granted that pain is a subjective experience-. In our study we pilot-tested the preliminary list of items with a group of 23 children and adolescents to ensure proper understanding and suitability, as a way to address the subjectivity in studying the construct.   

The originality of the paper is limited by their methodological orientation (this means that there are some methodological limitations for the research, but in terms of the paper, the core logic is strong).

Authors’ response: Again, we thank the reviewer for the kind comment.

This is why I recommend to introduce another qualitative instruments like focus groups or discussion groups, and over all, intersubjective analysis, but not for improving the paper but research.

Authors’ response: We not only will include these strategies in our future studies, we have also added a comment onto the Discussion to suggest these strategies for future studies (page 9, line 277-279).

“In addition, future studies should explore the contribution of combining the COPAQ with qualitative intersubjective analysis for assessing changes in the conceptualization of a child’s concept of pain.”

Reviewer 2 Report

In this paper, a measure of a child’s concept of pain, called SOCOPA, was proposed. It is claimed that SOCOPA is easy to administer and respond to. Basically, this study seems to be proposed to complement of COPI, that is a work of [39]. This study has both big advantage and weakness. The advantage is that it is a very simple and easy-to-understand measure that is easy for teenagers to respond to. On the other hand, a weakness is that, as the authors acknowledged, the survey was conducted on a small number of participants and was limited to specific areas of Spain. Nevertheless, supplement the following, considering that the research has more advantages than weaknesses.

First, describe in more detail how 324 participants were recruited and why the Tarragona region was chosen in Section 2.
Second, describe as much research as possible from other countries other than Spain in related research(in Section 2).
Third, describe in detail the advantages and disadvantages of COPI and SOCOPA (in this study, only the disadvantages of COPI are described) in Section 4 Discussion. Remember that each has both advantages and disadvantages.
Fourth, describe the future research challenges of this study, i.e. how you will expand this study in Section 5 Conclusions.

Author Response

In this paper, a measure of a child’s concept of pain, called SOCOPA, was proposed. It is claimed that SOCOPA is easy to administer and respond to. Basically, this study seems to be proposed to complement of COPI, that is a work of [39].

 Authors’ response: This study was not conducted to complement COPI, but to provide a better questionnaire to evaluate the concept of pain in children and adolescents.

This study has both big advantage and weakness. The advantage is that it is a very simple and easy-to-understand measure that is easy for teenagers to respond to.

Authors’ response: We thank the reviewer for the kind comments.

On the other hand, a weakness is that, as the authors acknowledged, the survey was conducted on a small number of participants and was limited to specific areas of Spain. Nevertheless, supplement the following, considering that the research has more advantages than weaknesses.

First, describe in more detail how 324 participants were recruited and why the Tarragona region was chosen in Section 2.

Authors’ response: Done as requested (see pages 3 and 5).

We have added (page 3, line 96):

“....a school large enough to have a sufficient sample and near to the School of Medicine and Health Sciences.”

And (page 5, line 186):

“Of the 556 possible participating schoolchildren, 324 agreed to participate.”

Second, describe as much research as possible from other countries other than Spain in related research (in Section 2).

Authors’ response: Done as requested. We have added new studies in Section 2 (see page 5, lines 138-141). The new references: (1) Linacre JM. Optimizing rating scale category effectiveness. J Appl Meas. 2002;3(1):85–106. Epub 2002/05/09. J Appl Meas. 2002;3(1):85-106; and (2) Linacre JM. Investigating rating scale category utility. J Outcome Meas. 1999;3(2):103-22. And also in section 1.

And in section 1, we have added this sentence (Page 3, line 67-69).

“Louw et al. [38] used this questionnaire as the assessment tool for one programme of education in the neurophysiology of pain in children aged 12 years old”.

We have changed the 30 reference (Spanish research) by (page 11, line 370)

“Nijs, J., Roussel, N., Paul van Wilgen, C., Köke, A., Smeets, R. Thinking beyond muscles and joints: therapists' and patients' attitudes and beliefs regarding chronic musculoskeletal pain are key to applying effective treatment. Man Ther 2013, 18(2), 96–102. https://doi.org/10.1016/j.math.2012.11.001”

Third, describe in detail the advantages and disadvantages of COPI and SOCOPA (in this study, only the disadvantages of COPI are described) in Section 4 Discussion. Remember that each has both advantages and disadvantages.

Authors’ response: We agree that questionnaires have positive and negative characteristics. This study was conducted to help overcome the limitations of the only questionnaire available that assesses how children and adolescents conceptualize pain. We describe the problems we thought were important to address, and how we approached them. In the new version we have added information and edited the text for clarity (see pages 2 and 9).

We have added this sentence (page 3, line 73-74):

“The COPI has fourteen items, all of which were developed according to contemporary pain science, and review from a panel of clinical and research-focused international pediatric pain experts.”

And on page 9, line 272:

“Future research should use COPAQ with other samples of schoolchildren or in clinical settings to explain the findings, as COPI did.

Fourth, describe the future research challenges of this study, i.e. how you will expand this study in Section 5 Conclusions.

Authors’ response: Done as requested. We have added this sentence in Conclusions (page 9: line 285).

“We encourage future investigations to combine the COPAQ with qualitative analysis for assessing changes in the child’s concept of pain”.

Reviewer 3 Report

In this study, the author developed a new questionnaire to assess a child’s concept of pain and (2) conduct a preliminary analysis to evaluate its psychometric properties.

Major revision

  1. Title is "Development of the Survey of the Concept of Pain", however, it is unclear whether your study develop the survey or questionnaire. Please reconsider the study title in the appropriate way.
  2. Line 84, your study was cross-sectional, two-phase, mixed-method study, please explain more about two-phase, mixed-method in detail.
  3. Line 129 Please calculate the sample size for the development of the questionnaire.
  4. Please reconstruct the followings way, such as construct validity, item difficulty, Item discrimination, internal reliability, test-retest reliability.

Author Response

Title is "Development of the Survey of the Concept of Pain", however, it is unclear whether your study develop the survey or questionnaire. Please reconsider the study title in the appropriate way.

 Authors’ response: The reviewer is right; the objective of the study was to develop a new questionnaire to assess how children and adolescents conceptualize pain, The title indicated that we developed the survey and also showed the name that we gave to this questionnaire (“Survey of the Concept of Pain “). We have reconsidered the title, and changed it to:

         “Development of the Conceptualization of Pain Questionnaire: a measure to study how children conceptualize pain”.

We have changed the name of the questionnaire to:

            “Conceptualization of Pain Questionnaire (COPAQ).”

Moreover, we have made all the changes in the manuscript accordingly.

.Line 84, your study was cross-sectional, two-phase, mixed-method study; please explain more about two-phase, mixed-method in detail.

Authors’ response: Yes, that is correct. The study was two-phase and the methods are different in each. The first one was a qualitative, Questionnaire Development and the second one was a quantitative, Psychometric Assessment.

To clarify that, we have divided the procedures into two phases in the manuscript (page 4, line 108 and line 127)

.“Phase 1: Content Development using Qualitative Methods.

Phase 2: Psychometric Assessment of the Questionnaire using Rasch Analysis”.

Line 129 Please calculate the sample size for the development of the questionnaire.

Authors’ response: In the original submission, we provided this information on lines 150-1, page 5. We realize that it would be better to move this information and description to the previous section (in the corresponding line 129, which now is lines 137-142, page 5):

“Sample size range was established on the basis that at least 10 observations per response option are needed for item threshold analysis [41] and at least 50 participants are needed to determine item fit with the Rasch model [42]. So, a sample of at least 150 participants should be available. Allowing for the removal of extreme scores, a sample of 243 persons is required for Rasch analysis to ensure item calibration stability within 0.5 logits with 99% confidence [34].

We have also added two new citations and references:

Linacre JM. Optimizing rating    scale category effectiveness. J Appl Meas. 2002;3(1):85–106. Epub 2002/05/09

Linacre JM. Investigating rating scale category utility. J Outcome Meas. 1999; 3(2):103–22

Please reconstruct the followings way, such as construct validity, item difficulty, Item discrimination, internal reliability, test-retest reliability.

Authors’ response: In this research, we have studied some of the psychometric characteristics of the newly developed measure, but not all of them. In our original submission, we highlighted this as a limitation, and mentioned that future studies would have to address the other characteristics. In the new version, we have added the characteristics that the reviewer is mentioning here (see page 9). In addition, we have edited the text for clarity (see pages 6 and 7).

  • Construct validity. Rasch analysis examines the extent to which items in a scale measure the same underlying construct. Construct validity is not evalued in our work, because when the questionnaire was developed there was no other measure of child’s “concept of pain”. The SOCOPA (now, COPAQ) construct validity derives from the item development.

We have added the following sentence (pag 4, line 110):

“in order to cover the seven dimensions that an expert survey on the concept of pain had identified”

And (page 9, line 276):

construct validity,

  • Item difficulty. Rasch analysis does not calculate the proportion who responded to an item correctly, which would evaluate an item's difficulty. Rasch analysis makes it possible to study the level of difficulty of the items using the Beta coefficient. Rasch theory states that the parameters should be interpreted as the ease with which each item is completed (the beta value of the model) so zero is the average value. Positive values indicate a difficult question and negative values indicate a simple question. In general, the cut-off points for determining whether an item is a poor fit are +3 and -3. In our study none of the Beta coefficients exceeded those cut-offs so no items were removed. Items 2, 5, 7, 8 and 15 could be considered difficult. The remaining items had a negative Beta coefficient, which indicates that they were easier to answer.

You are right, that is no enough clear, so we added in the manuscript, page 6, line 197.

“3.2.2. Unidimensionality (item difficulty)”

  • Item discrimination

We have added this as a limitation (page 9, line 276)

item discrimination

  • Internal reliability. Cronbach’s alpha is a measure used to assess the reliability, or internal consistency, of a set of scale or test items. In our particular case, we have described this in the statistical analysis: to measure internal consistency or reliability, Rasch analysis estimates the Person Separation Index (PSI), which is equivalent to Cronbach’s alpha in other types of analysis.

We agree that this is not clear enough, so we have added, page 7, line 232.

3.2.5. Internal Consistency or Reliability

Test-retest reliability. We did not give the questionnaire twice to the same people at different times to see if the scores were the same, so that is a limitation of our work. Consequently, we have added (page 9, line 276)

“…some important psychometric properties were not examined, including sensitivity to change over time, construct validity, item discrimination and test-retest reability”

Round 2

Reviewer 3 Report

I confirmed that this draft version has no major issues for publication.